# Content, Structure and Delivery Characteristics of Yoga Interventions for Managing Rheumatoid Arthritis: A Systematic Review Protocol

**DOI:** 10.3390/ijerph19106102

**Published:** 2022-05-17

**Authors:** Isha Biswas, Sarah Lewis, Kaushik Chattopadhyay

**Affiliations:** 1Lifespan and Population Health Academic Unit, School of Medicine, University of Nottingham, Nottingham NG5 1PB, UK; sarah.lewis@nottingham.ac.uk (S.L.); kaushik.chattopadhyay@nottingham.ac.uk (K.C.); 2The Nottingham Centre for Evidence-Based Healthcare—A JBI Centre of Excellence, Nottingham NG5 1PB, UK

**Keywords:** rheumatoid arthritis, management, yoga, systematic review

## Abstract

The global burden of rheumatoid arthritis among adults is rising. Yoga might be a potential solution for managing rheumatoid arthritis. This systematic review aims to synthesise the content, structure and delivery characteristics of effective yoga interventions for managing rheumatoid arthritis. The JBI methodology for systematic reviews of effectiveness and the Preferred Reporting Items for Systematic Reviews and Meta-analyses (PRISMA) guidelines will be followed. PRISMA for systematic review protocols (PRISMA-P) was used to write the protocol. Randomised controlled trials assessing the effectiveness of yoga interventions for managing rheumatoid arthritis in adults will be included in this review. We aim to search the following databases to find published and unpublished studies: ABIM, AMED, AYUSH Research Portal, CAM-QUEST, CINAHL, CENTRAL, EMBASE, MEDLINE, PeDro, PsycInfo, SPORTDiscus, TRIP, Web of Science, DART-Europe-e-theses portal, EthOS, OpenGrey and ProQuest Dissertations and Theses. No date or language restrictions will be applied. A narrative synthesis will be conducted. Meta-regression will be conducted to explore the statistical evidence for which components (content, structure and delivery characteristics) of yoga interventions are effective.

## 1. Introduction

### 1.1. Rheumatoid Arthritis

Rheumatoid arthritis is a chronic inflammatory autoimmune disease in which the immune system of the body damages the joints by attacking their membrane linings (synovium) and eroding the bones, cartilages, tendons and ligaments [1]. The main symptoms include warm, swollen, painful and stiff joints (particularly morning stiffness or after a period of inactivity) and additional ones such as poor appetite, weight loss, weakness, high fever and sweating [2,3]. It commonly occurs in the joints of the wrists, hands and feet [1,3]. The symptoms are usually first noticed in the small joints in the middle of the fingers and at the base of the fingers and toes, and sometimes in the shoulders, elbows, knees and ankles [1,3]. It is a symmetrical disease which means the symptoms usually occur on both sides of the body at the same time and to the same extent [1,3]. In this disease, there are unpredictable episodes of symptoms, from not feeling any symptoms to times when symptoms worsen known as flares [3]. These episodes vary from person to person, making it difficult to define flares [3]. This makes it difficult to differentiate between early and established rheumatoid arthritis [4]. Biologically, the most appropriate way of defining an established disease is when the diagnosis is confirmed [4]. Diagnosis is performed using physical examination based on symptoms and signs (e.g., inflamed tear glands or rheumatoid nodules under the skin) to assess the swelling of joints and their movement [3,5]. Main blood tests to confirm the diagnosis include erythrocyte sedimentation rate (ESR), c-reactive protein (CRP) and full blood count [3,5]. Other blood tests include measuring rheumatoid factors and anti-cyclic citrullinated peptides (anti-CCP) [3]. These are antibodies that are produced by the immune system when it attacks healthy tissue [1,3]. Those who test positive for both these factors are at increased risk of severe disease [3]. In some cases, joint scans such as X-ray, magnetic resonance imaging (MRI) and ultrasonography are also used [1,5]. Disease activity is an important outcome in rheumatoid arthritis and is used to understand the current state of the disease, get an idea of its progression over time and make treatment-related decisions [6].

### 1.2. Possible Causes and Risk Factors of Rheumatoid Arthritis

It is not clear why the immune system attacks the joints; however, there are some possible causes and risk factors of the disease [3,7]. These include sociodemographic factors such as increasing age and female sex; environmental factors such as smoking and outdoor air pollution; genetic factors such as having a family history of rheumatoid arthritis and altered genotype of the human leukocyte antigen gene complex (HLA-DRB1) that increases susceptibility to the disease; health conditions such as altered microbiome of the gut and obesity (body mass index (BMI) greater than 30 kg/m^2^) and hormonal changes especially in women (e.g., the onset of menopause) [7,8,9,10]. Some triggers, such as a period of physical or mental trauma or a period after a long illness, are also risk factors for rheumatoid arthritis [7].

### 1.3. Burden of Rheumatoid Arthritis

The global prevalence of rheumatoid arthritis in 2019 was 0.46% (95% confidence interval (CI): 0.39 to 0.54) [11]. It has substantial health (physical and psychological), social and economic burden [1,12,13]. Physical health consequences in the early stages of the disease include the inflammation of tear glands and salivary glands which reduces the production of tears and saliva and weak, deformed and stiff joints of the hands [1]. Major damage to the joints, Carpal Tunnel syndrome, inflammation of lungs and cardiovascular diseases such as myocardial infarction and stroke can occur in severe cases [1,3]. Depression is the major psychological health consequence of the disease [12]. Social implications include withdrawal from social life and limitations in performing activities of daily living [13]. The economic consequences include disease-related absenteeism from work and loss of work productivity leading to a reduction in income and an increase in treatment-related costs [13]. Overall, it has a negative impact on health-related quality of life [14]. In terms of numbers and percentages, rheumatoid arthritis resulted in 3.26 million global disability-adjusted life years (DALYs) in 2019 and was responsible for 0.1% of total global DALYs [15]. The global disability-adjusted life years (DALYs) related to rheumatoid arthritis are high and rising [15].

### 1.4. Current Management Strategies of Rheumatoid Arthritis and Their Limitations

There is no cure for rheumatoid arthritis, but the disease can be managed using pharmacological and non-pharmacological interventions [3]. The main aim is early diagnosis and treatment to prevent irreversible damage to the joints by lowering disease activity, reducing pain and inflammation and improving function [3,16]. Pharmacological interventions include the use of disease-modifying anti-rheumatic drugs (DMARDs) that are administered orally, subcutaneously or intravenously to suppress autoimmune activity and delay or prevent joint degeneration [16,17]. Conventional synthetic DMARDs (csDMARDs) such as methotrexate are used as first-line therapy due to their effectiveness and safety, flexible administration and low cost [17]. However, csDMARDs can have side effects such as sore mouth, loss of appetite, feeling sick and impacts on the gastrointestinal, hepatic, hematologic and pulmonary systems [18]. Biologic DMARDs (bDMARDs) such as infliximab are usually prescribed when csDMARDs are ineffective [19]. These are highly specific in their mechanism of action and can have serious adverse effects such as increased risk of opportunistic and bacterial infections and reactivation of tuberculosis [19]. Other pharmacological interventions include Janus kinase (JAK) inhibitors for people with severe disease and non-steroidal anti-inflammatory drugs (NSAIDs) (e.g., ibuprofen) to reduce pain by decreasing inflammation but can have side effects such as gastrointestinal ulceration, renal failure and heart failure [3,16]. Glucocorticoids (e.g., dexamethasone) are more effective but less safe than NSAIDs due to long-term side effects such as weight gain, muscle weakness and bone thinning [20]. In short, the medications used to treat rheumatoid arthritis can have strong side effects and are often not well-tolerated and ultimately can lead to high treatment costs [16]. Along with pharmacological interventions, non-pharmacological interventions such as brisk walking and moderate-to-high intensity exercise are recommended to improve or maintain mobility and joint function [1,21]. Joint surgery is recommended only at the end-stage of the disease, and joint replacements might lead to systematic articular infections [1,3].

### 1.5. Yoga: A Potential Solution for the Management of Rheumatoid Arthritis

Yoga is an ancient practice, with origins in the Indian subcontinent, that aims to offer a holistic sense of well-being of the body and mind [22]. Yoga philosophy and practice were first described by Patanjali in the classic text—*Yoga Sutras* [23]. The multi-factorial approach of yoga includes components such as yogic poses (asana), breathing practices (pranayama) and dhyana (meditation) and relaxation practices along with moderation in lifestyle [23]. Among the six major branches of yoga, Hatha yoga is the most popular [24]. There are various styles of Hatha yoga, and each has its distinct emphasis on the individual components [24]. Yoga is becoming increasingly popular across the world as there are about 300 million people who practice yoga globally [25]. Generally, yoga uses a gentle approach, is easy to learn and safe to practice, demands a low to moderate level of supervision, is inexpensive to maintain because of minimal equipment required and can be practised indoors or outdoors [26,27]. Yoga could be beneficial for a wide range of health conditions such as COVID-19, lung diseases, type 2 diabetes, hypertension, obesity, cardiovascular diseases, cancers, musculoskeletal conditions and mental health disorders [28,29,30,31,32,33,34,35,36,37].

Several systematic reviews have reported the beneficial effects of yoga interventions on rheumatoid arthritis outcomes, such as lower disease activity, pain relief and functional improvement [38,39,40,41,42]. These reviews have included randomised controlled trials (RCTs) except two which also included other study designs [38,41]. A recent systematic review and meta-analysis of 10 RCTs on knee, hand and feet rheumatoid arthritis showed that yoga significantly lowered disease activity (4 RCTs; standardised mean difference (SMD) −0.38; 95% CI: −0.71 to −0.06) and improved physical function (5 RCTs; SMD −0.32, 95% CI −0.58 to −0.05) compared to no intervention or usual care [42].

The positive effects of yoga on rheumatoid arthritis outcomes can be explained by some potential mechanisms. The varying postures in yoga (such as flexion, extension, adduction, abduction and rotation) strengthen and stabilise muscles by engaging them in contraction [43]. Strengthening of muscles leads to a reduction in pain and improvement in function [43,44]. Stress is a major psychological health factor that can lead to muscle tension and continued joint pain [45]. The continued pain reduces pain tolerance over time, disturbs emotional regulation and can further increase stress [46]. This can lead to increased activity of the sympathetic nervous system which stimulates the secretion of inflammatory biomarkers such as interleukin-6 (IL-6) causing inflammation [47]. The breathing techniques in yoga strike a balance between activity of the sympathetic and parasympathetic nervous system and counter the stress response, thus, reducing inflammation [48].

### 1.6. The Rationale for the Systematic Review

None of the above-mentioned systematic reviews has synthesised the content, structure and delivery characteristics of yoga interventions for managing rheumatoid arthritis [38,39,40,41,42]. Additionally, meta-regression has never been conducted to explore the statistical evidence for which components of yoga interventions (e.g., content, structure and delivery characteristics) are effective. Thus, there is a need to conduct such a systematic review so that the content, structure and delivery characteristics of effective yoga interventions for managing rheumatoid arthritis can be synthesised and used in future research and practice. For example, based on the findings, a yoga intervention could be developed, evaluated and implemented for managing rheumatoid arthritis.

### 1.7. Aim

The systematic review aims to synthesise the content, structure and delivery characteristics of effective yoga interventions for managing rheumatoid arthritis.

## 2. Methods

This systematic review will be conducted by following the JBI methodology for systematic reviews of effectiveness and the Preferred Reporting Items for Systematic Reviews and Meta-analyses (PRISMA) guidelines [49,50]. PRISMA for systematic review protocols (PRISMA-P) was used to write the protocol [51]. The protocol is registered with PROSPERO (CRD42022320337).

### 2.1. Inclusion Criteria

#### 2.1.1. Population

This systematic review will include studies conducted among adults (aged ≥ 18 years) diagnosed with rheumatoid arthritis in one or more joints. No restrictions will be applied regarding the diagnostic criteria of rheumatoid arthritis, including diagnosis based on physical examination, blood tests and joint scans. If a study includes both children and adults, only the relevant information about adults will be extracted. If it is not possible to extract the relevant information about adults, the study will be excluded.

#### 2.1.2. Intervention

Studies reporting at least one of the major components of yoga, namely, asana (yogic poses), pranayama (breathing practices) and dhyana (meditation) and relaxation practices, will be included. There will be no restrictions on the type, frequency, duration or delivery mode of the yoga intervention. Studies on multimodal interventions (which include yoga among other interventions) will be excluded if relevant data cannot be extracted. Studies will also be excluded if they did not explicitly label the intervention as yoga.

#### 2.1.3. Comparator

Studies comparing a yoga intervention with no intervention, sham intervention, non-pharmaceutical intervention (e.g., physical activity) or pharmaceutical intervention will be included. Studies having co-interventions will be included as long as all the eligible study groups were allowed to do so. Studies with a head-to-head comparison of two or more yoga interventions (i.e., different in terms of content, structure or delivery characteristics) will be excluded.

#### 2.1.4. Outcome

This systematic review will include studies that assessed the main outcomes of rheumatoid arthritis, i.e., disease activity, pain and function [52,53,54]. Disease activity measured using any of the validated composite disease activity scores (DAS) such as DAS28-ESR and DAS28-CRP, pain assessed using any scale such as the visual analogue scale (VAS) and numeric rating scale (NRS), and function assessed using any scale such as arthritis impact measurement scale (AIMS) and health assessment questionnaire (HAQ) will be eligible [54]. Radiographic outcomes such as structural joint damage and residual inflammation in the synovium will not be included in this review as inadequate blinding may affect the specificity of the classification criteria and are not standardised to be considered as core outcomes [52].

#### 2.1.5. Study Design

Considering the feasibility and practicality of the proposed work and the hierarchy of study designs, only RCTs will be included in this systematic review.

### 2.2. Data Sources and Search Strategy

The following 13 databases will be searched from their inception dates to find published studies—(i) A Bibliography of Indian Medicine (ABIM) (http://indianmedicine.eldoc.ub.rug.nl/, accessed on 14 April 2022), (ii) Allied and Complementary Medicine (AMED) (from 1985; Ovid), (iii) AYUSH Research Portal (http://ayushportal.nic.in/, accessed on 14 April 2022), (iv) CAM-QUEST (https://www.cam-quest.org/en, accessed on 14 April 2022), (v) CINAHL (from 1994; EBSCOHost), (vi) Cochrane Central Register of Controlled Trials (CENTRAL) (from 1996), (vii) EMBASE (from 1974; Ovid), (viii) MEDLINE (from 1946; Ovid), (ix) Physiotherapy Evidence Database (PeDro) (from 1999), (x) PsycInfo (from 1806; Ovid), (xi) SPORTDiscus (from 2004; EBSCOhost), (xii) Turning Research Into Practice (TRIP) (from 2014) and (xiii) Web of Science (from 1900; Clarivate analytics). Unpublished studies will be searched using—(i) DART-Europe-e-theses portal (from 1999), (ii) EthOS (from 1925), (iii) OpenGrey (from 1997) and (iv) ProQuest Dissertations and Theses (from 1980). The reference list of all the included studies and relevant previous systematic reviews will be screened for additional studies.

The search strategies are developed based on the following and in consultation with a research librarian at the University of Nottingham: (i) the yoga component is based on a relevant systematic review [55], (ii) the rheumatoid arthritis component is based on the search strategies reported in the UK’s National Institute for Health and Care Excellence (NICE) guideline on rheumatoid arthritis [56] and a Cochrane systematic review on rheumatoid arthritis [57], and (iii) the pre-designed search filters for RCTs are used [58,59,60]. All the search strategies are detailed in the Appendix A. No date or language restrictions will be applied.

### 2.3. Study Screening and Selection

All the identified citations will be collated and uploaded onto Endnote X9 (Clarivate Analytics, PA, USA) [61], and duplicates will be removed. The remaining records will be then imported into Rayyan (Qatar Computing Research Institute [Data Analytics], Doha, Qatar) to facilitate the title and abstract screening process [62]. Titles and abstracts will be independently screened for eligibility using the inclusion criteria by two systematic reviewers (IB and SL/KC). Studies identified as potentially eligible or those without an abstract will have their full text retrieved. Full texts of the studies will be assessed for eligibility by two independent reviewers. Full-text studies that do not meet the inclusion criteria will be excluded, citing reasons. Any disagreements that arise between the two reviewers will be resolved through discussion. If consensus is not reached, a third reviewer (SL/KC) will be consulted. Translations will be sought where necessary. 

### 2.4. Assessment of Methodological Quality

Included studies will be critically assessed by two independent systematic reviewers (IB and SL/KC) for methodological quality using the standardised critical appraisal tool developed by JBI for RCTs [49]. This tool uses a series of criteria that can be scored as being met (yes), not met (no), unclear or not applicable (n/a). The two reviewers will independently assess each criterion and comment on it. Any disagreements that arise between the two reviewers will be resolved through discussion. If consensus is not reached, then a third reviewer (SL/KC) will be involved. All studies, regardless of their methodological quality, will undergo data extraction and synthesis, where possible.

### 2.5. Data Extraction

Two systematic reviewers (IB and SL/KC) will independently extract data from the included studies using a pre-developed and pre-tested data extraction form. Any disagreements that arise between the two reviewers will be resolved through discussion. If consensus is not reached, a third reviewer (SL/KC) will be consulted. The following data will be extracted: author(s), year of publication, country, participant characteristics (e.g., age, sex, ethnicity, occupation, type of joint affected by rheumatoid arthritis), sample size, intervention and comparator, outcomes, the timing of follow-up at the end of the intervention and adverse events. For all the outcomes, the authors will extract the end of intervention data [52,57]. Where this time point is not reported, data from the time point closest to the end of intervention will be extracted. Intention-to-treat (ITT) data will be preferred compared to per-protocol data. ITT analysis is the most preferred analysis method for RCTs [63]. This is because it preserves sample size by including all the participants irrespective of adherence to the study and attrition, maximises external validity and helps in understanding real-world circumstances encountered in an actual setting [63,64]. Post-intervention data will be extracted in preference to change from baseline data (i.e., post-intervention score—baseline score). Percentage change from baseline will not be extracted as it is highly sensitive to change in variance and it also fails to protect from baseline imbalances, leading to non-normally distributed outcome data [65]. In addition, the content of yoga interventions will be extracted (e.g., yogic poses, breathing practices, meditation and relaxation practices) along with the structure (e.g., duration of the yoga sessions, duration and frequency of the yoga interventions) and delivery characteristics (e.g., individual or group sessions, supervised or unsupervised sessions, sessions delivered in yoga centres or at home, strategies for yoga intervention uptake and adherence, characteristics of yoga instructors).

To obtain missing data on outcomes, multiple strategies will be used. The first strategy will be to contact the corresponding author of the included study by email (at least two times per author) to obtain the relevant data. If we get no response from the corresponding author, then certain assumptions will be applied. For example, where disease activity, pain and function are reported as continuous outcomes, if the standard deviation (SD) is missing, SD will be imputed from a similar study (in terms of intervention, comparator, sample size and numerical outcome data) [66]. If only a median and interquartile range (IQR) is reported, these will be extracted, the mean will be assumed to be equal to the median and the SD will be calculated using the standard formula (=IQR/1.35) [66].

### 2.6. Data Synthesis

A narrative synthesis will be conducted with the help of tables, focusing on the content, structure and delivery characteristics of effective yoga interventions for managing rheumatoid arthritis (i.e., for each type of joint and outcome). For example, in the case of rheumatoid arthritis of the wrist, a narrative synthesis will be performed for disease activity, pain and function of the wrist.

Considering the errors in how authors analyse and report yoga interventions to be effective in studies (e.g., doing pre–post analysis of outcomes within study arms but no comparative analysis between study arms), meta-analyses will be conducted for each type of joint and outcome using Review Manager 5.4.1 (Copenhagen, The Nordic Cochrane Centre, The Cochrane Collaboration) [67] to determine the true effectiveness of each included yoga intervention. Meta-analyses based on random-effects models will be conducted to provide a weighted measure of treatment effect. The Ι^2^ statistic will be used to quantify statistical heterogeneity across studies. For studies with more than one comparator group, the comparisons will be included in separate meta-analysis models to avoid the issue of double-counting of the comparator group. Where disease activity, pain and function are reported as continuous outcomes, mean differences (MDs) with 95% CIs will be reported where the same scale is used across the studies. Where different scales are used across studies, SMDs with 95% CIs will be reported. Where necessary, post-intervention data will be pooled with changes from baseline data, and this will be conducted only for MDs but not SMDs. If reported as binary outcomes, risk ratios with 95% CIs will be reported. Where there is a sufficient number of studies (at least 10) included in a meta-analysis, a funnel plot will be generated to assess for publication bias.

In the final step, meta-regression will be conducted to explore the statistical evidence for which components of the intervention are effective [68]. This requires a reasonable number of studies to have sufficient enough power to show differences in effectiveness between components, and so the final decision on which components will be explored and how they will be grouped will be made once we have extracted data from the included studies on the components of each intervention. However, we anticipate exploring broad categories of intervention components including content (e.g., yogic poses, breathing exercises, meditation and relaxation practices), structure (e.g., number of yoga sessions per week, length of the yoga sessions) and delivery characteristics (e.g., one-to-one or group sessions) of yoga interventions [68]. A random-effects model will be used to analyse these subgroup effects [69]. The effects of content, structure and delivery components on outcomes, i.e., disease activity, pain and function, will be investigated by looking at the amount of heterogeneity explained by these components using the reduction in the I^2^ statistic and the DerSimonian–Laird estimation method [70]. The results will establish the statistical significance of any observed patterns in which components of yoga are associated with a greater effect on rheumatoid arthritis outcomes.

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
