# Peer review of "Content, Structure and Delivery Characteristics of Yoga Interventions for Managing Rheumatoid Arthritis: A Systematic Review Protocol"

_ijerph, 2022, doi:10.3390/ijerph19106102_

Round 1

Reviewer 1 Report

The authors need to address the feasibility aspect of the study in the submitted protocol. Otherwise, the draft is fine.

The protocol: „ Content, structure and delivery characteristics of yoga interventions for managing rheumatoid arthritis: a systematic review protocol" by Isha Biswas and colleagues propose a randomised controlled trial assessing the effectiveness of yoga interventions for managing rheumatoid arthritis in adults. Yoga, the art of physical, mental and spiritual practices, has been widely practiced in the Indian subcontinent for ages but the practice is not exclusively accepted in the western world. However, in recent times, the spectrum of different forms of Yoga has been spreading globally. The submitted protocol addressing yoga intervention for managing rheumatoid arthritis seems interesting. This will signify the healing aspects of Yoga. The preliminary protocol is impressive but I feel that  the authors  need to address the feasibility aspect of the study, especially, how many studies have been approximately published addressing the concerned issue and whether the data of all the studies could be analysed. This is important to point out the conclusions based on statistical outcome. Secondly, the authors need to briefly address the therapeutical importance of the study.

Reviewer 2 Report

srma = (systematic review and meta-analysis)

dear colleagues, 

thank you so much for the srma protocol. some suggestions:

1)databases need to be ordered alphabetically. 

2)it should be primsa-p (protocol version)

3)introduction did not provide logic for srma. it was merely narrative review. revise lines 121-126 too to make clear case of srma yoga+ra. 

a meta-analysis statistician should be consulted subgroup meta-analysis and meta-regression are mainly ways to explore heterogeneity. 

4)since this is JBI I assume one author is JBI collab + should be registered there as well beside prospero. 

5)pico is better in table and prior inclusion/exclusion criteria. 

6)inclusion/exclusion criteria not clear. provide clear lists. 

7)line 218 Any disagreements that arise between the two reviewers will be 
resolved through discussion - add after discussion with the third author. 

8)data synthesis plan is very basic and offers basic template-based analyses authors need to add i2, tau, tau2, and h to heterogeneity and also gosh and influence analyses these can be done using python or R. many interactions can be assessed and several advanced arithmetics can be added. 

9)provide some hypotheses and discuss these 

10) also add point 16/17 http://www.prisma-statement.org/documents/PRISMA-P-checklist.pdf 

minor

a) prisma flow diagram is missing 

b) consider updating to primsa2020

Reviewer 3 Report

The manuscript is well structured and I have only a few concerns

L54-63 Paragraph not strictly necessary

Instead, I would expand the paragraph on Yoga and its clinical applications and implication, not only to rheumatic pathology:
Pranayama in COVID-19 https://link.springer.com/article/10.1007/s12070-020-01883-0 ; COPD https://pubmed.ncbi.nlm.nih.gov/33221632/
Asana for Back Pain https://doi.org/10.1016/j.jaim.2021.01.011

This does enrich the rationale for the holistic approach of the intervention

Perhaps the outcome section is a bit simplistic, I would help suggesting this broader scheme ... https://doi.org/10.1016/j.berh.2019.101482
Indeed: Joint range of motion (goniometry)? Grip strength? WHODAS II ? SF-36, NHP, EuroQoL, RAQoL, Patient Global Assessment, SF-12, SF-6D ? etc etc

L224 Why not ROB2?
